# Flower-Visiting Insects Ensure Coffee Yield and Quality

Jesús Hernando Gómez [1], Pablo Benavides [1], Juan Diego Maldonado [1], Juliana Jaramillo [2], Flor Edith Acevedo [3] and Zulma Nancy Gil [1,*]

[1] Entomology Department, Centro Nacional de Investigaciones de Café—Cenicafé, Planalto Headquarters, km. 4 vía Antigua a Manizales, Manizales 170009, Caldas, Colombia; jhgomezl@unal.edu.co (J.H.G.); pablo.benavides@cafedecolombia.com (P.B.); juandiegomaldonadoc@gmail.com (J.D.M.)

[2] Theme Lead Regenerative Agriculture, Rainforest Alliance, De Ruyterkade 6 BG, 1013 Amsterdam, The Netherlands; jjaramillo@ra.org

[3] Department of Entomology, Pennsylvania State University, Penn State Behrend Campus, 651 Cemetery Rd, North East, PA 16428, USA; fea5007@psu.edu

* Correspondence: zulma.gil@cafedecolombia.com

**Abstract:** (1) Background: The participation of insects in the pollination of self-pollinating plants, such as coffee, is still controversial. This study determined the effect of flower-visiting insects on coffee berry set, yield, and quality. (2) Methods: Over 2 years, five evaluations in different locations, dates, and harvest times were carried out. Each evaluation consisted of eight treatments with 50 replicates each, arranged in a completely randomized experimental design. Treatments were established to identify the contribution of insects, wind, gravity, self-pollination, and cross-pollination to coffee yield and quality. (3) Results: The insects contributed 16.3% of the berry set, 26.9% of the berry coffee yield, and 30.6% of the weight of supremo-type beans. No differences were observed in the sensory quality of coffee produced with or without insects. For stigma receptivity, results indicate that there is a 6.3% probability of self-pollination during pre-anthesis. (4) Conclusions: The species *Coffea arabica*, despite being a self-pollinating plant, benefits from the presence of flower-visiting insects. During anthesis, arabica coffee flowers are ready for cross-pollination.

**Keywords:** coffee; *Coffea arabica*; berry set; insects; flower visitors; coffee yield; coffee quality





## 1. Introduction

Pollination in plants is defined as the process by which pollen is carried from the stamens (male sexual organs) to the stigma (female sexual organ), giving rise to fertilization and later to the formation of berries [1]. There are two main types of pollination: Cross-pollination or xenogamy, which occurs when pollen originates from another plant, and self-fertilization, also called autogamy, which occurs when pollen originates from the same plant. Xenogamy is more successful than autogamy, since it avoids inbreeding and produces greater genotypic and phenotypic variability in natural populations [2,3]. A vector is needed for cross-pollination to occur. According to Willmer [1], this vector can be abiotic (wind, gravity, or water) or biotic (animals commonly called pollinators, such as insects, birds, reptiles, or mammals). Of the latter, insects are among the main vectors. About 67% of flowering plants are pollinated by insects, which explains why they are considered the most important pollinators in both wild and cultivated plants [4,5]. Among insects, bees stand out for being strictly flower visitors that use nectar and pollen as source of food [6]. Other studies also highlight the importance of bees in the pollination of agricultural systems and their effect as flower visitors increasing quality, fruit set, and yield in crops [7–9].

For example, some species of tomato [*Solanum* sp. L. (Solanaceae)], pumpkin [*Cucurbita* sp. L. (Cucurbitaceae)], and some species of passion flowers [*Passiflora* sp. L. (Passifloraceae)], are self-incompatible, and thus depend on cross-pollination [10–12]. Although species, such as avocado [*Persea americana* Mill (Lauraceae)], cotton [*Gossypium* ssp. (Malvaceae)], and coffee [*Coffea arabica* L. (Rubiaceae)] are self-pollinated, they also benefit

from the pollination service provided by insects with an increased yield and larger fruit or berry size [10,13–16]. Among these crops, coffee has been widely studied regarding the effect of pollinators on yield, mainly in *Coffea arabica* and *Coffea canephora* Pierre ex. A. Froehner (Rubiaceae). The species *C. canephora* is self-incompatible, and thus depends on cross-pollination for berry yield, mainly mediated by insects with bees being the most important [17]. *Coffea arabica*, on the other hand, is a species whose flowers are self-pollinated. However, several studies report that bee-mediated pollination increases production [13,18–21].

Studies on *C. arabica* variety Caturra KMC® Amaral [18,22] found that, when branches were left exposed to visitation by insects, these accounted for 13.6–39.2% of the increase in berry set as compared with branches excluded from pollinators. Sein [23] found 60% of berry set in flowers protected from insects as compared with 70% in exposed flowers. Subsequently, Badilla and Ramirez [24] found a 15.85% increase in berry set in *C. arabica* variety Catuí rojo, which was attributed to pollination by insects. In a study carried out by Roubik [25] in Panama, the bee *Apis mellifera* L. (Hymenoptera: Apidae) was found to increase *C. arabica* production by 50%, whereas in a study conducted by Klein et al. [19] in Indonesia, results of treatments where coffee was exposed to the presence of bees differed significantly from those of treatments excluding bees. The 12.3% increase in berry set was attributed to the presence of bees. Other studies also found that coffee berries presented a higher weight when flowers were exposed to pollinators [25–28]. Even improved cup quality in terms of enhanced flavor and aroma characteristic has been attributed to pollination by bees [29]. In some cases, the contribution of insects turns to be less than 10% as in Colombia, the first study that addressed the role of flower-visiting insects in coffee crops that was carried out by Castillo [30] on *C. arabica* variety Cera. Study results indicated that the proportion of berries derived from insect-mediated pollination remained below 10%, rarely exceeding this value and never surpassing 20%. Planting distances and varying sample size, however, were found to affect study results. In the aforementioned study, exclusion treatments consisted of placing flowers in paper bags, which modified temperature and humidity. The data obtained corresponded to an index and not to an exact measurement of cross-pollination frequency. Although Arcila [31] mentioned that 90% of self-pollination in *C. arabica* occurs in the pre-anthesis stage, this information lacks experimental support. An exploratory study conducted by Jaramillo [21] found that insect-mediated pollination in this same coffee species helped in reducing the number of aborted berries and contributed to larger berry size and higher sugar concentration (degrees Brix), which could improve coffee quality. Finally, in a study conducted by Bravo-Monroy et al. [32] in the province of Santander (Colombia), a 10.5% increase in berry set was reported when flowers were exposed to pollinators. Although many of the aforementioned studies had a small sample size and limited repeatability, their results are still relevant and give an indication of the effects of insect-mediated pollination in coffee crops.

Whether insects contribute to coffee production has been a controversial issue for many years due to the plant's autogamy. Therefore, this study aims to determine the effect of flower-visiting insects on the percentage of coffee berry set, production, and quality, based on the hypothesis that the visits made by insects to coffee flowers account for more than 10% of berry set and production while also improving quality. The probability that coffee flowers self-pollinate during pre-anthesis was determined. To achieve the objective, two locations in the center of the coffee-growing region of Colombia were selected. For 2 years (i.e., five flowering events), eight treatments with 50 replicates each were evaluated in a complete randomized design.

## 2. Materials and Methods

### 2.1. Study Sites

The study was carried out at two experiment stations of Colombia's National Coffee Research Center (Cenicafé, for its Spanish acronym) located in the country's central coffee-

growing region: The Naranjal Experiment Station and the La Catalina Experiment Station (Table 1, Figure 1).

**Table 1.** Description of study sites located in Colombia's central coffee-growing region.

| Study Sites | Location | Climate Conditions | Harvest Distribution | References |
|---|---|---|---|---|
| Naranjal Experiment Station | Municipality of Chinchiná (department of Caldas), Cordillera Central Mountain range, western slope (4°58′ N, 75°39′ W); 1381 m above sea level; coffee ecotope: 206 A. | Average temperature, 21.6 °C; average relative humidity, 80.6%; annual precipitation, 2990 mm; 1537 h of sunshine/year. | Main harvest (75%): Flowering between January and March; mid harvest (25%): Flowering between August and September | [31] |
| La Catalina Experiment Station | Municipality of Pereira (department of Risaralda), Cordillera Central Mountain range, western slope (4°45′ N, 75°44′ W); 1321 m above sea level; coffee ecotope: 218 A. | Average temperature, 22.1 °C; average relative humidity, 78.9%; annual precipitation, 2464 mm; 1588 h of sunshine/year. | Main harvest (75%): Flowering between January and March; mid harvest (25%): Flowering between August and September | [31] |

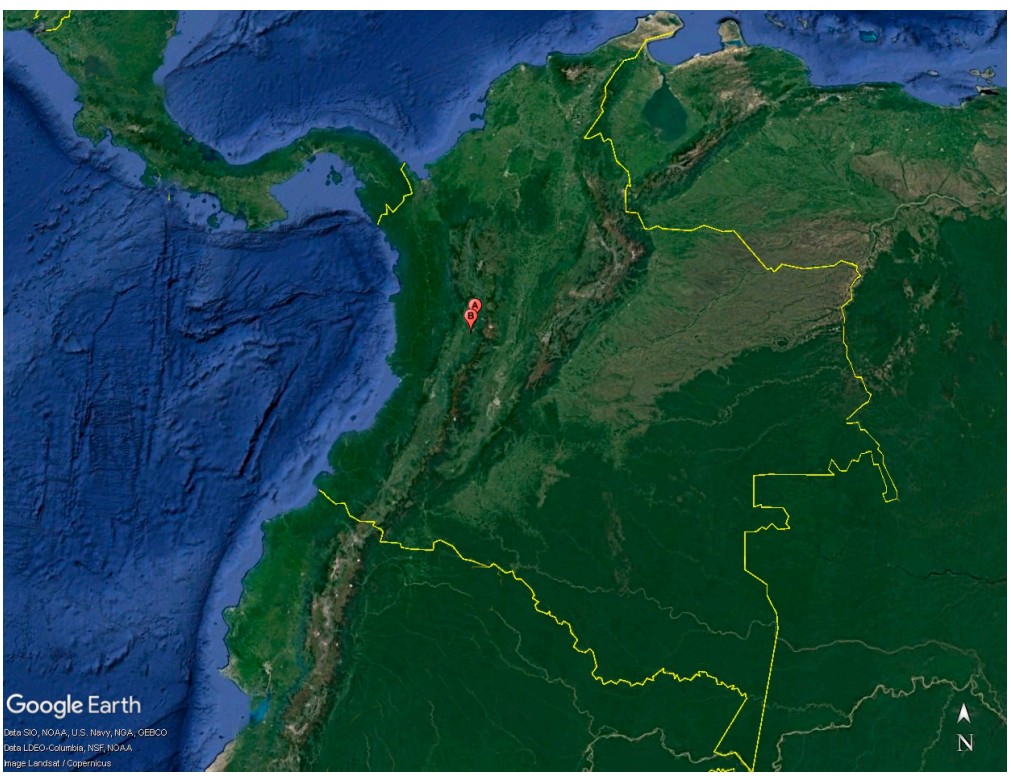

**Figure 1.** Study sites located in Colombia's central coffee-growing region. (A) Naranjal Experiment Station and (B) La Catalina Experiment Station.

*2.2. Effect of Flower-Visiting Insects on Coffee Berry Set, Yield, and Berry Physical Quality*

Five independent evaluations were carried out between 2019 and 2021 to determine the effect of flower-visiting insects on percentage of coffee berry set, yield (total weight of cherry coffee berries), and physical quality (total weight of supremo-type beans). Three of the evaluations were conducted in the main harvest flowering which occurs in the first semester of the year, and correspond to 65% of the harvest and two in mid-crop flowering which occurs in the second semester of the year and correspond to 35% of the harvest (Table A1). Five plots of sun-grown *C. arabica* variety Castillo® in reproductive stage (2–3 years old), were selected in the study area. Each plot measured a minimum of 0.5-hectares, with a planting density of 5000 plants per hectare.

For each evaluation, 400 coffee trees were randomly selected from each plot and randomly assigned to the following treatments (T): T1: designed to determine the effect of pollination without the participation of insects, wind, gravity, and self-pollination, where coffee branches were enclosed in sleeves of mesh fabric that allowed for the entrance of pollen but not insects; T2: designed to determine the effect of wind and gravity on coffee flowers pollination, with coffee branches emasculated and enclosed in sleeves of mesh fabric; T3: evaluated the effect of self-pollination (unemasculated flowers) and consisted of coffee branches enclosed in sleeves made with 300-thread-count fabric that did not allow for the entrance of pollen or insects; T4: evaluated the natural pollination with participation of insects and consisted of coffee branches fully exposed (not enclosed); T5: evaluated the effect of wind, gravity, and insect pollination on coffee flowers, had coffee branches emasculated and exposed (not enclosed); T6: a control for emasculation, consisted of coffee branches emasculated and enclosed in 300-thread-count fabric that did not allow for the entrance of pollen and insects; T7: determined the effect of cross-pollination, branches were emasculated, hand pollinated with pollen from other plants, and enclosed in sleeves of 300-thread-count fabric that did not allow for the entrance of pollen and insects; T8: evaluated the effect of self-pollination, branches were emasculated, hand pollinated using pollen from the same plant, and enclosed in sleeves made of 300-thread-count fabric that did not allow for the entrance of pollen and insects (Tables 2 and A2).

**Table 2.** Graphic representation of the treatments. (+) = treatment applied; (−) = treatment not applied.

| | Types of Pollination | | | | | |
|---|---|---|---|---|---|---|
| **Treatment (T)** | **Self-Fertilization** | **Wind** | **Gravity** | **Insects** | **Manual Pollinatation Using Pollen from the Same Plant** | **Manual Pollination Using Pollen from Other Plants** |
| T1 | + | + | + | − | − | − |
| T2 | − | + | + | − | − | − |
| T3 | + | − | − | − | − | − |
| T4 | + | + | + | + | − | − |
| T5 | − | + | + | + | − | − |
| T6 | − | − | − | − | − | − |
| T7 | − | − | − | − | − | + |
| T8 | − | − | − | − | + | − |

Fifty randomly distributed experimental units, each consisting of one tree, were evaluated per treatment.

From each tree, one reproductive branch with at least 30 flower buds was selected prior to applying the treatments. Only those buds in the pre-anthesis stage were left on the branch and counted, while those buds that where in a different stage were eliminated, such as berries and flower primordia. The buds that did not comply with this characteristic were discarded.

Two types of entomological sleeves were used, which differ in the type of fabric and diameter spacing between threads, both types of white color, with dimensions of 89.0 cm long X 29.0 cm wide. (1) Sleeves made of mesh fabric with spacing between threads of 0.5–0.7 mm that allow for the passing of coffee pollen grains that are approximately 0.035 mm (±0.007 mm n: 6) in diameter (Figure 2); (2) sleeves made of 300-thread-count fabric with a maximum spacing between threads of 0.01 mm that do not allow for the passing of coffee pollen grains (Figure 2). The entomological sleeves were removed 8 days after the treatments were applied. Treated branches were subsequently cleaned every 15 days by eliminating newer forming flower buds. At 90 days after flowering, the number of berries set per treatment was counted and the percentage of berry set was calculated as the ratio of the number of formed berries to the initial flower buds.

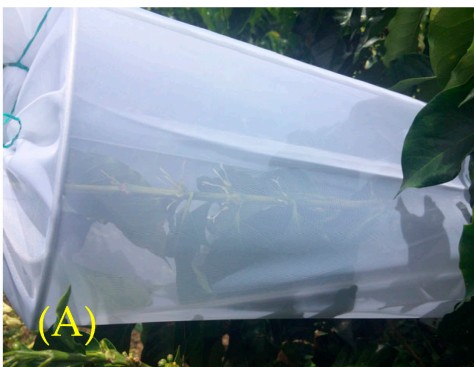
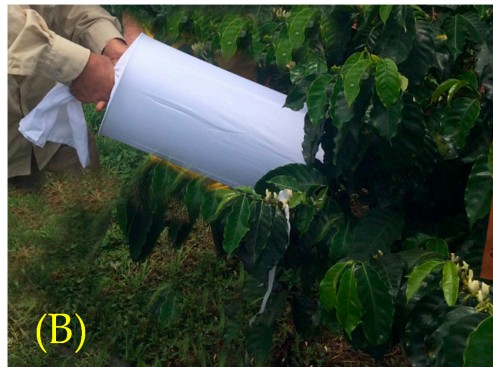

**Figure 2.** Types of entomological sleeves. (**A**) Sleeves made of mesh fabric with spacing between threads of 0.5–0.7 mm and (**B**) sleeves made of 300-thread-count fabric with a maximum spacing between threads of 0.01 mm.

Thirty-two weeks after flowering, ripe coffee berries were harvested from each treated branch to determine the weight of coffee berries as a measure of yield. Several harvesting passes were made to ensure that berries were picked at the same stage of maturity. The coffee berries were then processed (pulped, demucilaginated, mechanically dried, and threshed) to evaluate the weight of supremo-type beans as a measure of the physical quality of the coffee bean using 431.8/1625.6 mm circular screens [33].

### 2.3. Effect of Flower-Visiting Insects on Coffee's Sensory Quality

To evaluate the effect of flower-visiting insects on coffee's sensory quality, a 0.5-hectare plot planted to *C. arabica* variety Castillo® was selected at the Naranjal Experiment Station. A total of 250 trees were randomly selected and distributed into five experimental units, each with 50 trees. Four branches with flower buds in the pre-anthesis stage were randomly selected from each tree. Of these, two branches were covered with entomological sleeves to prevent the passing of insects, while the other two branches were tagged and left exposed to floral visitors. The sleeves were removed 1 week later and ripe coffee was harvested between weeks 32 and 36. The harvested coffee of each experimental unit was processed, roasted, and ground under homogeneous conditions. The sensory quality in cup was estimated based on the SCA scoring (Specialty Coffee Association). The SCA score has a scale from 0 to 100 and only coffees with more than 80 points are considered as a specialty [34].

### 2.4. Stigma Receptivity and Presence of Pollen during Pre-Anthesis

A 0.5-hectare plot planted with 3-year-old *C. arabica* variety Castillo® in flowering stage was selected at the Naranjal Experiment Station to evaluate the stigma receptivity of coffee flowers and presence of pollen during pre-anthesis.

To evaluate stigma receptivity, the hand cross-pollination method was used [35]. Forty trees were randomly selected to have 10 trees for each flowering stage (pre-anthesis, anthesis, day 1 after anthesis, day 2 after anthesis). A branch with at least 10 flower buds in pre-anthesis was selected from each tree, individual buds at different stages of development and fruits were removed. Each flower bud was emasculated and branches were enclosed in a 300-thread-count fabric sleeve to prevent the entry of pollen and insects. At each flowering stage described above, flowers were cross-pollinated by hand, started the same day at the afternoon to day 2 after anthesis. After 24 h, stigmas were removed and fixed on FAA (10% formaldehyde, 50% alcohol at 96%, 5% acetic acid, and 35% distilled water).

A total of 329 stigmas were analyzed, of which 63 were in the pre-anthesis stage, 104 in the anthesis stage, 103 were on day 1 after anthesis stage, and 59 on day 2 after anthesis stage. The stigmas were transferred to a solution of 0.6% NaOH solution at a temperature of 60 °C for 20 min, and subsequently, the stigmas were washed five times in distilled water. Pollen tubes were stained with aniline blue, and then observed under a Nikon Eclipse

90i fluorescence microscope and counted for each flower stage. Stigmas were considered receptive when the pollen tubes surpassed half of the stigmas length [36]. Based on this information, the proportion of receptive stigmas with their respective standard error was estimated for each flowering stage.

Ten trees were randomly selected from the same plot to evaluate the presence of pollen in anthers. One branch with flower buds in the pre-anthesis stage was chosen from each tree and eight flowers were removed from each branch, four at 10:00 a.m. and four at 3:00 p.m. The pollen was immediately collected from each flower using glycerinated gelatin stained with acid fuchsin. The pollen samples in the glycerinated gelatin were placed on slides and observed under a Carl Zeiss Primo Star light microscope. The proportion of samples with presence of pollen was determined.

*2.5. Statistical Analysis*

For each evaluation and treatment, the average and standard error were estimated for the following variables: Percentage of berry set, yield, and physical quality. Analysis of variance was applied in all evaluations using a completely randomized experimental design with results showing the effect of the different treatments. In the case of the variable percentage of berry set, Dunnett's multiple comparison test was applied at 5% to compare the T6 treatment (control) with the other treatments and is used when we want to compare one group (usually the control treatment) with the other groups. Subsequently, analysis of variance was applied excluding T6 and the Tukey test was applied at 5%. The aforementioned analyses were applied to all the information obtained regardless of the evaluation. In the case of the variables for yield and physical quality, the Duncan's multiple range test was applied at 5% to determine differences between treatments. It was used to compare all pairs of means, excluding the control treatment (T6). All these analyses were performed using the statistical software SAS 9.4 [37].

Data obtained in T4 and T1 were used as basis to obtain the absolute relative difference with respect to T4 Equation (1), which served to determine the effect of insects on percentage of berry set as insects were present in T4 but were excluded from T1. This same absolute relative difference was estimated for the variables of yield and physical quality.

$$\frac{T_4 - T_1}{T_4} x100 \tag{1}$$

A qualitative analysis was performed based on the SCA scoring to determine the effect of flower-visiting insects on the sensory quality of coffee in cup.

For the analysis of stigmatic receptivity (variable of interest), an analysis of generalized linear models at 5% was performed, with a binomial response distribution, using the R software version 3.5.0 [38], using the package "MCMCglmm" version 2.33 [39], to estimate the stigmatic receptivity in each floral stage (i.e., pre-anthesis, and days 1 and 2 after anthesis) in terms of probability, along with the determination of differences between floral stages under the multiple comparison with Tukey test adjustment at 5%.

To determine the presence of pollen in the anthers, an analysis of generalized linear models at 5% was performed, with a binomial response distribution, using the R studio software using the package "MCMCglmm" version 2.33 [39]. Subsequently, the probability of presenting stigmatic receptivity and pollen at the same time was estimated, and in this way, the probability that a flower self-pollinates in pre-anthesis.

## 3. Results

*3.1. Effect of Flower-Visiting Insects on Coffee Berry Set, Yield, and Berry Physical Quality*

For response variables "Percentage of berry set" and "Yield" as well as for the complementary variable "Physical quality", the analysis of variance showed an effect of treatments with >95% reliability. Dunnett´s multiple comparison test at 5% showed statistical differences between T6 (control for emasculation) and the rest of the treatments for the variable percentage of berry set. T6 was accordingly excluded from the Tukey test at 5%. Moreover,

this treatment presented the lowest values for percentage of berry set, yield, and physical quality (Table 3). This trend was maintained across the five evaluations (Table 4), indicating that the emasculation technique was well performed.

**Table 3.** Averages and standard error (SE) for the response variables "Percentage of berry set", "yield", and "physical quality".

| Treatment (T) | Percentage of Berry Set | | Yield | | Physical Quality | |
|---|---|---|---|---|---|---|
| | Average (%) | SE | Average (g) | SE | Average (g) | SE |
| T1 | 69.4 C [a] | 0.9 | 68.2 B [b] | 2 | 10.2 B [b] | 0.5 |
| T2 | 13.2 F | 0.7 | 14.3 E | 2.3 | 2.0 D | 0.2 |
| T3 | 74.1 B | 0.8 | 66.9 B | 2 | 10.1 B | 0.4 |
| T4 | 82.9 A | 0.6 | 93.3 A | 2 | 14.7 A | 0.6 |
| T5 | 60.2 E | 1.5 | 58.8 C | 2 | 8.9 BC | 0.5 |
| T6 | 3.2 | 0.4 | 6.4 F | 3.7 | 0.9 D | 0.2 |
| T7 | 66.0 DC | 1.05 | 51.1 D | 2.1 | 7.5 C | 0.3 |
| T8 | 61.8 DE | 1.2 | 51.0 D | 2.1 | 7.2 C | 0.3 |
| Percentage effect [c] | 16.30% | | 26.90% | | 30.60% | |

[a] Different letters indicate a difference between averages according to the Tukey test at 5%. [b] Different letters indicate a difference between averages according to the Duncan's multiple range test at 5%. [c] Absolute relative difference between T4 and T1.

**Table 4.** Average and standard error (SE) for the response variable "percentage of berry set" for the five evaluations.

| Treatments (T) | Evaluation 1 | | Evaluation 2 | | Evaluation 3 | | Evaluation 4 | | Evaluation 5 | |
|---|---|---|---|---|---|---|---|---|---|---|
| | Average | SE | Average | SE | Average | SE | Average | SE | Average | SE |
| T1 | 56.4 B [a] | 3 | 66.0 C | 1.6 | 75.3 B | 1.2 | 72.4 B | 2 | 76.8 B | 1.6 |
| T2 | 3.5 | 0.7 | 9.1 | 1 | 8.3 D | 1 | 17.9 D | 1.4 | 27.1 D | 1.7 |
| T3 | 71.8 A | 2.3 | 74.9 B | 1.9 | 74.0 B | 1.2 | 72.5 B | 2.2 | 77.1 B | 2.2 |
| T4 | 77.0 A | 1.9 | 82.4 A | 1.3 | 86.1 A | 0.8 | 83.9 A | 1.1 | 84.8 A | 1 |
| T5 | 31.0 C | 2.4 | 44.7 D | 2.9 | 71.2 C | 2.1 | 72.8 B | 1.7 | 81.3 BA | 1.4 |
| T6 [b] | 0.5 | 0.2 | 4.1 | 1.9 | 0.06 | 0 | 6.8 | 1.6 | 4.3 | 0.8 |
| T7 | 72.4 A | 2.8 | 71.0 C | 1.8 | 63.9 C | 1.2 | 58.4 C | 2.9 | 63.7 C | 1.9 |
| T8 | 50.1 B | 2.9 | 73.6 B | 1.9 | 72.4 B | 2 | 50.1 C | 2.7 | 62.5 C | 2.5 |
| Percentage effect [c] | 26.7 | | 19.9 | | 12.5 | | 13.7 | | 9.4 | |

[a] Different letters indicate a difference between averages according to the Tukey test at 5%. [b] Excluded from the analysis. [c] Absolute relative difference between T4 and T1.

For the variable percentage of berry set, when averaged over the five evaluations, T4 (natural pollination with insect participation) differed statistically from the others, presenting a higher percentage of berry set of $82.9 \pm 0.6\%$ (Table 3), ranging between $77.0 \pm 1.9\%$ and $86.1 \pm 0.8\%$ (Table 4), whereas T1 (exclusion of insects) presented a berry set of $69.4 \pm 0.9\%$ (Table 3), and ranged between $56.4 \pm 3\%$ and $76.8 \pm 1.6\%$ (Table 4).

The treatment T3 presented the maximum values of autogamy, which averaged $74.1 \pm 0.8\%$ (Table 3). Across the five evaluations, autogamy ranged between $71.8 \pm 2.3\%$ and $77.1 \pm 2.2\%$ (Table 4).

The treatment T5 involved emasculation, as a result, self-pollination was absent in anthesis. Wind, gravity, and insects alone accounted for $60.2 \pm 1.5\%$, on average, of the percentage of berry set (Table 3) and, across evaluations, from $31.0 \pm 2.4\%$ to $81.3 \pm 1.4\%$ (Table 4).

The average percentage of berry set in T7 was $66 \pm 1.5\%$ and in T8 was $61.8 \pm 1.2\%$ (Table 3). Across evaluations, these values ranged from $58.4 \pm 2.9\%$ to $72.4 \pm 2.8\%$ for T7 and from $50.1 \pm 2.7\%$ to $73.6 \pm 1.9\%$ for T8 (Table 4).

The treatment T2 where only wind and gravity participated in pollination contributed on average 13.2 ± 0.7% of the percentage of berry set (Table 3). Across evaluations, this value ranged between 3.5 ± 0.7% and 27.1 ± 1.7% (Table 4).

For the variable weight of cherry coffee berries, T4 also differed statistically from the rest of the treatments according to the Duncan's multiple range test at 5%, presenting a higher weight of 93.3 ± 2.0 g, on average, whereas T1 presented an average weight of 68.2 ± 2.0 g (Table 3).

For the complementary variable physical quality of coffee, the Duncan's multiple range test at 5% also showed the effect of treatments. Once again, T4 differed statistically from the others, presenting the highest physical quality with an average of 14.7 ± 0.6 g (Table 3). The treatment T1 presented an average weight of 10.2 ± 0.5 g.

For the variable percentage of berry set, the absolute relative difference between T4 and T1 was, on average, 16.3% (Table 3) and, across evaluations, ranged between 9.4% and 26.7% (Table 4). In the case of the yield variable, the absolute relative difference was 26.9% and, in the case of the physical quality, 30.6% (Table 3).

*3.2. Effect of Flower-Visiting Insects on Coffee's Sensory Quality*

According to the SCA grading sheet, scores were between 81.25 and 83.63 points for both treatments where branches were protected from insects as well as the one where they were exposed to insects. This score corresponds to a specialty coffee graded as premium coffee (Table 5).

**Table 5.** SCA scores for each of the treatments where scores between 81.25 and 83.63 points are graded as premium coffee.

| Repetition | SCA Scores | |
|---|---|---|
| | **Insects Exposed Branches** | **Branches Covered with Entomological Sleeves (Exclusion of Insects)** |
| 1 | 81.88 | 81.25 |
| 2 | 83.06 | 81.56 |
| 3 | 82.63 | 83.63 |
| 4 | 82.63 | 83.63 |
| 5 | 82.63 | 81.75 |

*3.3. Stigma Receptivity and Presence of Pollen during Pre-Anthesis*

Stigmas were clearly visualized and pollen tubes were observed. Analysis showed statistical differences, with the pre-anthesis stage presenting the lowest stigma receptivity at a probability of 50.7% ± 0.253. The estimated probabilities of stigma receptivity were not different among the other three flowering stages ($p > 0.05$), although the estimates for days 1 and 2 after anthesis were both as high as around 98% (Table A3). The probability of the presence of pollen at 10:00 a.m. and 3:00 p.m. was 15.0% and 10.0%, respectively, and there were no differences between the evaluated timings (df = 0.049, z = 0.6883, $p > 0.05$). However, the value probability of presence of pollen in pre-anthesis was 12.5% and the probability of occurrence of self-pollination in pre-anthesis was 6.3%.

**4. Discussion**

Little importance has been given to studies on cross-pollination in coffee since the cultivated species *C. arabica* is self-compatible. However, this study found that the contribution of flower-visiting insects to coffee berry set was greater than 10% in four of the five evaluations carried out, which is higher than the percentage (10%) found by Castillo [30], even for the same locality (Naranjal Experiment Station).

The present study also found that the average participation of insects in coffee berry set was 16.3%, reaching values of 26.7%. These percentages are within the ranges of 10–30% reported by other researchers [19,22–24,40–43].

These results are relevant since they change the present state of knowledge regarding arabica coffee pollination and coffee berry set. It seems that this was underestimated in previous studies on the topic, e.g., the ones by Castillo [30] and Bravo-Monroy et al. [32].

The contribution of flower-visiting insects to coffee berry set varied across the five evaluations, which can be attributed to different biotic and abiotic conditions present during the evaluations, such as climatic variables [44] and phenological aspects of the flower and abundance of flowering [44,45]. The proximity of certain coffee plots to forest remnants or conserved areas was also observed to have an effect and can lead to a greater abundance and richness of flower-visiting insects, which in turn is related to a higher percentage of berry set [46,47].

Another influencing factor is the seasonality of insects during the flowering period [45]. Several climatic variables are known to differentially affect the visits made by the different bee taxa. For some species, solar brightness is positively correlated with bees' foraging activity in coffee flowers, while for others it correlates negatively [44].

The percentage of berry set was statistically similar when flowers were emasculated and pollen came from another plant (T7) or from the same plant (T8), this may be due to the fact that the study plots were planted to improve *C. arabica* variety Castillo®, this variety is formed by 29 progenies developed from crosses of Caturra with the hybrid Timor. These lines are compatible with low genetic variability among plants. Furthermore, these lines were selected for their high fertility [48].

However, based on the results obtained, it cannot be concluded that cross-pollination in *C. arabica* results in a higher percentage of berry set as compared with self-fertilization. Results do show, however, that cross-pollination occurs in the coffee plant, which is in accordance with findings reported by other authors who found increases from 10 to 54% in percentage of berry set when cross-pollination occurred [19,49,50]. However, the descriptions of the methodology used in the aforementioned studies are not clear about whether flowers were emasculated or not. It is also important to mention that the emasculation technique could have reduced the percentage of berry set. Jimenez and Castillo [51] found that this technique reduced berry set values due to possible damage to the stigma, since it implies the removal of the entire corolla.

In studies conducted in Colombia, Castillo [30] and Jiménez and Castillo [51] reported a 90% contribution of self-pollination to berry set, whereas other studies reported a lower percentage contribution (29–47.9%) (e.g., [19,47]). The present study found that coffee berry set attributable to self-pollination averages around 74.1%, as observed in T3, even reaching a percentage of up to 77.1%. The treatment T3 was influenced by a high concentration of pollen within the experimental unit as branches were covered with entomological sleeves made of a 300-thread-count fabric, which avoided the removal of pollen by the action of wind and rain. On the contrary, T1 allowed pollen to circulate freely from the inside to the outside of the covered branch.

The variation of data between the different studies can be attributed to the different varieties used as coffee yield is known to vary depending on the hybridizations performed as part of the development process of new varieties [52,53]. In the same way, crop yield is affected by both the genotype and its interaction with the environment [54].

The present study found that the percentage values of berry set in T2 (13.2%), where wind and gravity pollination occurred, exceeded those reported by Castillo [30] of 7.8%. Although the effect of climate on pollen dispersal has not been exhaustively studied in *C. arabica*, Castillo [30] found that temperature and precipitation influence pollen dispersal, as temperature facilitates anther dehiscence. González et al. [55] found that rain serves to settle air-borne pollen. In the case of the present study, both temperature and precipitation varied across the five evaluations.Wind speed, land shape, and planting distance are also known to affect wind and gravity pollination [30].

The contribution of wind-, gravity-, and insect-mediated pollination to the percentage of berry set was 60.2% in the case of T5. This value indicates that, in the absence of self-pollination, primarily insects, followed in a lesser extent by wind and gravity, would

contribute significantly to the pollination of coffee flowers. The percentage found in T5 surpasses the 21.9% reported by Carvalho and Krug [50], but falls below the 93.47% reported by [21]. In the case of the present study, the factors already mentioned altered the contribution of wind and gravity to the percentage of berry set. In addition, although flowers were emasculated in this treatment, removing a large part of the corolla, some bees visited flowers searching for nectar. Studies conducted by Pierre et al. [56] showed that, even though bees do not have contact with the reproductive structures, they could help in dispersing pollen in the air if they fly close to the flowers, increasing pollination by wind and gravity, which in turn could have also influenced the values obtained in this study.

This study reports a contribution of flower-visiting insects to a yield of 26.9%. This proportion is within the ranges reported by Roubik [26] which was between 25 and 56%. Several studies have shown that bee-mediated pollination increases the weight of coffee berries and leads to better physical characteristics of berries [26,28,29]. In Jamaica, Raw and Free [57] conducted an experiment with *A. mellifera*, placing plants of *C. arabica* variety Caturra in cages. Berry weight doubled in plants placed in cages with bees present. In a subsequent study carried out by Roubik [26] in *C. arabica* varieties Caturra and Catimor, results showed that visits by bees increased the final retention of berries by 25%, in addition to a 56% increase in yield in the open pollination treatment as compared with the treatment in which flowers were enclosed and did not allow for insect contact with the flowers.

In an exploratory study carried out by Jaramillo [21] in Colombia, emasculated flowers exposed to pollinating agents (wind, gravity, and insects) presented a higher average weight per berry as well as larger berry size as compared with the rest of the evaluated treatments. These data, however, are not comparable with those reported in the present study since Jaramillo [21] evaluated the average weight per berry fruit and the present study evaluated the total weight of cherry coffee berries collected in each treatment. We believe that it could be an appropriate measure to determine the impact on coffee yield.

According to norms established by the Colombia's National Federation of Coffee Growers (FNC, its Spanish acronym)—National Committee of Coffee Growers [58], coffee bean size can be used to estimate yields, and thereby its final export price. Larger beans are regarded as having a better physical quality. Beans retained by a screen 18 (7.10 mm mesh opening) during classification are graded as premium, and those retained by a screen 17 (6.70 mm mesh opening) are graded as supremo. In this study, the total weight per treatment of supremo-type beans was evaluated as a variable of physical quality, with higher weights occurring in T4 (exposed to insects) as compared with T1 (exclusion of insects) and T3 (self-pollination). This suggests that flower-visiting insects affected the total weight of supremo-type beans by 30.6%.

The proportion of larger coffee beans in any given harvest may be attributed to visiting insects, but could also be related to genetic factors, which vary among varieties. This possibly explains why the proportion of supremo-type beans surpasses 80% in coffee variety Castillo$^{®}$ [48]. In this study, all treatments and experimental units involved the same varieties and environmental conditions, allowing for the effect of the treatments to be evaluated, finding statistically significant differences between treatments.

Regarding the effect of flower-visiting insects on the sensory quality of coffee, this study graded those treatments where branches were protected from insects in the same way as those where branches were exposed to insects. No treatment was graded above 85 points (considered excellent for specialty coffee). Similar studies report that pollination in the presence of insects improves cup quality, improving its flavor and aroma [29]. Many factors, however, determine coffee cup quality—from cultivation conditions, such as the presence of shade, altitude, temperature, and use of fertilizers, to crop genetics to processing and post-harvest practices [31,59–63].

According to Arcila [31], 90% of the flowers are self-pollinated in the pre-anthesis stage; however, this statement is not based on an experimental study. Studies carried out by Krug and Teixeira [64] and Cabrera [65] determined that the percentage of self-pollination in pre-anthesis can reach up to 10%. The present study determined that the probability of

self-pollination occurring in pre-anthesis is only 6.3%, a value ratified by T6 that serves as an indicator to determine the number of self-pollinated flowers in pre-anthesis. This percentage across the five evaluations was below 10%, which implies that the methodology applied is not only correct, but also that a large part of the crop is not pollinated in the pre-anthesis stage and that most of the pollination apparently occurs from anthesis onwards, when, according to Krug and Teixeira [64] and Alvim [66] pollen is released and the stigma is receptive. Although *C. arabica* flowers can last from 1 to 2 days after being pollinated [67,68], when pollination does not occur within this timeframe, they can last up to 5 days [68], which is a sufficient time window for insect-mediated pollination to occur in the crop. Finally, it was found that the species *Apis mellifera* L. (Hymenoptera: Apidae) was the one that most visited the flowers of *C. arabica* with 55.4% of the total visitors and 65.5% of the bees. Regarding the native bees, the presence of *Tetragonisca angustula* Latreille, 1825 (Hymenoptera: Apidae) stood out, being the second most abundant species with a representation of 16.4% of the sample of bees).

## 5. Conclusions

The *C. arabica* species, despite being a self-pollinated plant, benefits from the presence of flower-visiting insects in the crop, which contributed 16.3% to berry set, 26.9% to yield, and 30.6% to the physical quality of coffee. Although the effects of the presence of flower visitors in coffee have already been studied by other aforementioned authors, this study differs from the rest in that a larger number of evaluations were carried out. A total of 2000 experimental units were analyzed, making it statistically very robust. The study also measures new variables, such as the weight of supremo-type beans, which is important to determine the physical quality of coffee.

No sensory attributes linked to the presence of flower-visiting insects were found in *C. arabica*, possibly since this variable depends more on other factors that contribute to cup quality. It should be mentioned that in Colombia the premise was that *C. arabica* coffee was pollinated prior to anthesis [31]; however, this study showed that most of the pollination in *C. arabica* occurs from flower anthesis onwards, the time at which flower-visiting insects can perform the pollination service. Therefore, although arabica coffee is a self-pollinating plant, these results stress the importance of flower-visiting insects in contributing to the productivity and profitability of coffee.

**Author Contributions:** Fieldwork, data collection and analysis, manuscript writing and revision, J.H.G.; study design, methodological adjustments, data analysis, manuscript writing, P.B.; data collection and analysis, J.D.M.; experimental design, managed project funding, and revised the final version of the manuscript, J.J.; fieldwork, data collection, methodological adjustments, and revised the final version of the manuscript, F.E.A.; resource management, research coordination, conceptualization, methodology, fieldwork, data collection and analysis, manuscript writing and revision Z.N.G. All authors have read and agreed to the published version of the manuscript.

**Funding:** This research was funded by the collaborative research agreement CN-2017-1336 subscribed between Bayer AG, Germany (financing entity) and National Coffee Research Center-Cenicafé.

**Institutional Review Board Statement:** Not applicable.

**Data Availability Statement:** All data are contained within the article or Appendix A.

**Acknowledgments:** This study was part of the project "Evaluating flower-visiting insects in coffee in Colombia, with emphasis on bees, and their effect on yield and quality", under the collaborative research agreement CN-2017-1336 subscribed between Bayer AG, Germany (financing entity) and National Coffee Research Center - Cenicafé. We would like to thank Luis Eduardo Escobar for his support in experimental set up and data collection. We are also thankful for the 25+ field crew at the Naranjal and La Catalina Experiment Research Stations for their support with flower emasculation and set up of the different treatments.

**Conflicts of Interest:** The authors declare no conflict of interest. The funders had no roles in the design of the study; in the collection, analyses, or interpretation of data; in the writing of the manuscript, or in the decision to publish the results.

## Appendix A

**Table A1.** Evaluations carried out from 2019 to 2021.

| Evaluation Number | Date | Experiment Station | Experimental Plot | Type of Flowering |
|---|---|---|---|---|
| 1 | October 2019 | La Catalina | Pulmón | Mid-crop |
| 2 | January 2020 | Naranjal | Macadamia | Main |
| 3 | August 2020 | Naranjal | Maní | Mid-crop |
| 4 | February 2021 | Naranjal | Samana | Main |
| 5 | February 2021 | Naranjal | San José | Main |

**Table A2.** Description of treatments to test the effect of floral-visiting insects on coffee crop yield.

| Effect to Measure | | Treatment |
|---|---|---|
| Effect of wind and gravity on fertilization of coffee flowers | (1) | Coffee branches enclosed in bags with mesh openings that allow for the entrance of pollen from other coffee branches or trees, but do not allow for the entrance of insects. |
| | (2) | Emasculated coffee branches enclosed in bags with mesh holes that allow for the entrance of pollen from other coffee branches or trees, but do not allow for the entrance of insects. |
| Self-fertilization effect | (3) | Coffee branches enclosed in bags that do not allow for the entrance of pollen or insects. |
| Natural fertilization | (4) | Free branches (not enclosed). |
| Effect of wind, gravity, and insects on fertilization of coffee flowers | (5) | Emasculated free branches (not enclosed). |
| Control of the emasculation procedure | (6) | Emasculated coffee branches enclosed in bags with mesh openings that do not allow for the entrance of foreign pollen or insects. |
| Cross-pollination effect | (7) | Manual pollination with pollen from other coffee plants. Coffee branches are enclosed in bags that do not allow for the entrance of pollen or insects. |
| Manual self-fertilization effect | (8) | Manual pollinated branches using pollen from the same plant. Coffee branches are enclosed in bags that do not allow for the entrance of pollen or insects. |

**Table A3.** Averages and standard errors (SE) for the probability of stigma receptivity in different flower stages and probability of the presence of pollen in pre-anthesis at 10:00 a.m. and 3:00 p.m.

| Flower Stages | Probability of Stigma Receptivity | SE | t-Value [b] | p-Value [b] | Probability of the Presence of Pollen at 10:00 a.m. | Probability of the Presence of Pollen at 3:00 p.m. |
|---|---|---|---|---|---|---|
| Pre-anthesis | 50.7 B [a] | 0.253 | 0.125 | <0.001 | 15.0 A [b] | 10.0 A [b] |
| Anthesis | 88.0 A | 0.308 | 6.597 | <0.001 | | |
| Day 1 after anthesis | 98.0 A | 0.718 | 5.46 | <0.001 | | |
| Day 2 after anthesis | 98.0 A | 1.014 | 4.003 | <0.001 | | |

[a] Different letters indicate a difference between averages according to the Tukey test at 5%. [b] Not significant differences (df = 0.049, z = 0.6883, $p > 0.05$); the total value of probability of pollen presence in pre-anthesis was 12.5%.

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
