# Peer review of "Flower-Visiting Insects Ensure Coffee Yield and Quality"

_agriculture, doi:10.3390/agriculture13071392_

Round 1

Reviewer 1 Report

Dear authors,

The research is interesting, however some changes are needs

Author Response

Thanks for the valuable corrections to improve the paper. In the attached text you will find the responses to your corrections.

Reviewer 2 Report

Dear authors:

There is an inconsistency between the title and the Sensory Quality Findings in Table 4. Clearly, insect pollination did not improve quality; please revise in this regard.

It is evident that understanding the experimental design, which involves eight treatments, is crucial for comprehending the entire study. While the authors did provide a good explanation in the M&M section, I suggest a more reader-friendly way. I suggest that the authors create an additional table that presents a concise and easily understandable description of the treatments. This would also greatly aid in interpreting tables 2 and 3. See an example as the following:

Wind

Quality

Self-pollination

Insect

……

Note

T1

+

+

+

-

……

T2

+

+

-

-

……

T3

-

-

+

-

……

……

…..

…..

…..

…..

……

T8

…..

…..

…..

…..

……

Please also find additional comments and edits in the attached PDF.

Best wishes.

I suggest language editing to improve the expressions, mainly on the result part. 

Author Response

(The authors gave the same response as above.)

Reviewer 3 Report

This is a huge work, with a lot of experimentation on two sites and two years in difficult enviromnents, so I congratulate the researcher team to handle this job and I highly recommend this paper for a publication.

But, there is still minor change to bring to the ms and one major which consists to an improvement to the GLMM statistical analysis (see in details).

Details:

37-38: Check reference of Ollerton 2011 which indicates that it is more than 80% of plant which insect pollinated

56 : "were left exposed in plants" left exposed to what ? maybe "in plants" to remove

57: Add the publication date in brackets for Sein [23]

86-91: I would add a spatio-temporal context of your study as it is in 2 experiment stations in Colombia. 

94-97: Can you also add your experimental station on a map? It is always interesting for the readers to situate your study

112: Color and size of the meshes ? Ok I see it on 135 line

109-127: Can we also have a picture of each treatments ?

135: I do not understand this "Two types of entomological sleeves were used." please ca you clairify ?

160: Can you briefly explain what is the SCA scoring ?

191: Rename it by "Statistical analysis"

195-197: Why do you use the Dunnet's test, please add a reference for this

199-201 : Same question for Duncan test

211-218: Please specify what are your fixed and random effects, what is the variable to expalin, and why do you follow binomial distribution of errors, do you previously transform your data ?

You can check all of this here :  https://peerj.com/articles/4794/

283: As I am an entomologist researcher first, I would like to suggest if you can add a tiny paragraph in discussion part about the main animal pollinator for coffee crop in your region that could explain the highest value for T4.

412: Italic for "C. arabica" same for 421 line, please check it in all the text

Author Response

(The authors gave the same response as above.)

Round 2

Reviewer 3 Report

For me it is OK. Congrats to the authors.